# The Safety and Efficacy of 1-Monoeicosapentaenoin Isolated from the Trebouxiophyceae *Micractinium* on Anti-Wrinkle: A Split-Face Randomized, Double-Blind Placebo-Controlled Clinical Study

**DOI:** 10.3390/jcm12020587

**Published:** 2023-01-11

**Authors:** Ki Mo Kim, Kon-Young Ji, Yoon Jung Choi, Jong Beom Heo, Ui Joung Youn, Sanghee Kim, Ki-Shuk Shim, Joo Young Lee, Tae Soo Kim, Young Kyoung Seo, Gyu-Yong Song, Sungwook Chae

**Affiliations:** 1KM Convergence Research Division, Korea Institute of Oriental Medicine, 1672 Yuseongdae-ro, Yuseong-gu, Daejeon 34054, Republic of Korea; 2Korean Convergence Medicine, University of Science and Technology, 217 Gajeong-ro, Yuseong-gu, Daejeon 34113, Republic of Korea; 3College of Pharmacy, Chungnam National University, Daejeon 34134, Republic of Korea; 4Division of Life Sciences, Korea Polar Research Institute, Incheon 21990, Republic of Korea; 5Dermapro, Safety and Efficacy Evaluation of Cosmetics & Cosmeceutucals, Seocho-gu, Seoul 06684, Republic of Korea

**Keywords:** Trebouxiophyceae *Micractinium*, 1-monoeicosapentaenoin, clinical trial, skin wrinkle, anti-aging, matrix metalloproteinase

## Abstract

The skin aging process is governed by intrinsic and extrinsic factors causing skin wrinkles, sagging, and loosening. The 1-monoeicosapentaenoin (1-MEST) is a component isolated from *Micractinium*, a genus of microalgae (Chlorophyta, Trebouxiophyceae). However, the anti-wrinkle effects of 1-MEST are not yet known. This study aimed to evaluate the anti-wrinkle effects of 1-MEST in vitro and in clinical trials. The cytotoxicity of 1-MEST was investigated in vitro using the MTS assay in human epidermal keratinocytes (HEKs). Expression of matrix metalloproteinase (MMP)-1 and MMP-9 was determined by ELISA in HEKs irradiated with UVB after treatment with 1-MEST. A split-face randomized, double-blind, placebo-controlled study was conducted to evaluate the safety and efficacy of 1-MEST. The study evaluated wrinkle parameters and visual assessment, self-efficacy and usability questionnaires, and adverse events. The study showed that the 1-MEST was not cytotoxic in HEKs, suppressed MMP-1 secretion and MMP-9 protein expression in HEKs irradiated with UVB. The wrinkle parameters and mean visual assessment score were significantly decreased in the test group after 12 weeks and differed from the control group. There were no significant differences in efficacy and usability. Adverse effects were also not observed. The 1-MEST showed anti-wrinkle properties to slow down or prevent skin aging.

## 1. Introduction

The aging process is classified into two categories: intrinsic and extrinsic aging. The intrinsic aging is governed by genetic predisposition, whereas extrinsic aging is caused by radiation, stress, smoking, and pollution [1,2]. Skin aging is characterized by changes in skin thickness, structural changes in dermal and epidermal layers, and elastic fibers of the dermal layer [1,2]. The clinical features of skin aging include coarse, thin, lax skin accompanied by wrinkles, lentigines, irregular hyperpigmentation, telangiectasias, and sallowness [3]. People become more concerned about their skin condition as they age [4]. Skin aging is characterized by wrinkle formation accompanied by changes in the generation and breakdown of collagen [5]. A previous study reported that matrix metalloproteinase (MMP) production is induced by UVB exposure, which leads to increased extracellular matrix (ECM) degradation and wrinkle formation [6,7]. The secretion of MMPs from keratinocytes increasingly impairs collagen synthesis and the degradation of collagen and ECM proteins, leading to wrinkling and skin aging [6,7].

Drugs are generally prescribed and used according to the medical condition of patients but only for a limited duration [8]. Since cosmetics are generally used for several years, to inhibit wrinkle formation they should be certified to be safe [9]. It is also important to determine the safety of individual constituents used on the skin in various cosmetic products [10]. The safety of cosmetics is determined by experts to evaluate adverse events, including erythema, edema, swelling, and papules after they are applied to normal skin [11].

The Trebouxiophyceae *Micractinium* (*Micractinium* sp.) is a genus of microalgae (Chlorophyta) and a polyphyletic group of unicellular photosynthetic eukaryotes that comprises of many species of green algae [12,13]. *Micractinum* sp. used in the study was discovered as a new species on a sea ice on coast of the Baton Peninsula in the Antarctic, making it an excellent candidate for the discovery of new bioactive compounds (Appendix A). *Micractinum* sp. can live both sea and fresh water, and recent study has been reported that *Micractinium singularis* which is the closest species of *Micractinium* sp. was discovered living in sea water Janghang harbor, Korea [14]. To survive in such harsh environments, these microalgae have been reported to produce special compounds such as antifreeze proteins, polyunsaturated fatty acids, UV radiation-screening compounds, and antioxidants [15,16,17]. Among them, the anti-wrinkly effects of 1-monoeicosapentaenoin (1-MEST) have not yet been reported. To identify whether 1-MEST is a potential anti-wrinkle agent, we evaluated the effects of 1-MEST in human epidermal keratinocytes (HEKs) in vitro. Moreover, the safety and efficacy of 1-MEST were assessed for inhibiting wrinkle formation via clinical trials, and we confirmed the anti-wrinkle effects of 1-MEST and its potential use as an anti-wrinkle agent.

## 2. Materials and Methods

### 2.1. Preparation of 1-MEST

1-MEST was isolated from *Micractinium* sp. (KSF0031) as previously described [17]. However, the amount of isolated compound is not enough to perform in vitro and clinical evaluations. The synthesis process was performed as follows in the Figure 1. 1-ethyl-3-(3-dimethylaminopropyl)carbodiimide (EDC, 380 mg, 1.98 mmol, 1.2 equiv.) and 4-dimethylaminopyridine (4-DMAP, 40 mg, 0.33 mmol, 0.2 equiv.) were added to a solution of solketal (218 mg, 1.65 mmol, 1 equiv.) and eicosapentaenoic acid (500 mg, 1.65 mmol) in dry methylene chloride (MC, 30 mL). The reaction mixture was stirred at room temperature (RT) for 24 h. After stirring, the mixture was concentrated, and distilled water (50 mL) was added to it. Next, the mixture was extracted with Et_2_O (50 mL). The organic layer was washed with brine and dried over anhydrous Na_2_SO_4_. After concentration in vacuo, the residue was purified by flash column chromatography to obtain compound 1 (2,2-dimethyl-1,3-dioxolan-4-yl)methyl (5Z,8Z,11Z,14Z,17Z)-eicosa-5,8,11,14,17-pentaenoate) as a transparent liquid (542 mg, 78.8%). To a solution of compound 1 (542 mg, 1.30 mmol, 1 equiv.) in distilled water (1 mL), 4 mL acetic acid was added at 50 °C, and the reaction mixture was stirred for 5 h. Next, the mixture was cooled and neutralized with aqueous saturated NaHCO_3_ (15 mL). After adding 50 mL of distilled water, the solution was extracted with ethyl acetate. The organic layer was washed with brine and dried over Na_2_SO_4_ anhydrous. After concentrating in vacuo, the residue was purified by silica gel column chromatography to obtain compound 2 (5Z, 8Z, 11Z, 14Z, 17Z-eicosapentaenoic acid-2, 3-dihydroxypropyl ester) as a pale-yellow liquid (405 mg, 82.7%).

### 2.2. Analysis of MMP Suppressing Effect of 1-MEST In Vitro

We previously reported the detailed methods for in vitro analysis of MMP expression [18]. In brief, HEKs were purchased from Lonza (Walkersville, MD, USA) and maintained in KGM-Gold SingleQuots^TM^ medium (Lonza) containing growth factors and other supplements according to the manufacturer’s instructions. The cells were treated with 1-MEST for 24 h, and their viability was measured by MTS analysis using a CellTiter 96 Aqueous One Solution Cell Proliferation Assay (Promega, Madison, WI, USA). For UVB exposure, the HEKs were treated with 1-MEST at the indicated concentrations for 24 h and subsequently irradiated using a UVP Crosslinker (Ultra-Violet Products Ltd., Cambridge, UK) at 302 nm (UVB light; 20 mJ/cm^2^) for 1 min. The levels of MMP-1 secreted in culture supernatants were detected using the MMP-1 ELISA kit (R&D Systems, Minneapolis, MN, USA) as per the manufacturer’s instructions. The protein expression levels of MMP-9 were determined by Western blot analysis using primary antibodies against MMP-9 (#13667) and β-actin (#4970) and anti-rabbit secondary antibody (#7074); all antibodies were purchased from Cell Signaling Technology (Danvers, MA, USA). The signal intensities were measured using a Las-3000 image analyzer (Fujifilm, Tokyo, Japan) and quantified using ImageJ software (NIH, Bethesda, MD, USA).

### 2.3. Participants

The detailed methodology used in this study has been described earlier [19,20,21]. Briefly, we recruited over 20 volunteers who started or had already formed wrinkles and were aged between 30 and 65 years according to the guidelines for efficacy evaluation of functional cosmetics established by the Ministry of Food and Drug Safety (MFDS) in Korea. Finally, 24 volunteers aged 38 to 56 (average age 49.88 ± 4.34 years), who met the inclusion criteria, participated in this study (Table 1).

### 2.4. Preparation of 1-MEST Cream

The preparation of base of the cream is as described in our previous report [21]. The cream containing 0.1% 1-MEST (test cream) and the cream without 1-MEST (placebo cream) was used in this study as the test and control groups, respectively.

### 2.5. Double Blinding and Randomization

The test manager prepared the test products (placebo cream and test cream) by the same method using similar formulation ingredients and packed in the same container. To blind the test products, the test manager termed blind codes of the placebo cream and test cream such as “Product S” and “Product P”, respectively. In addition, the test manager provided the “Product S” and “Product P” to the investigators who were blinded to the identity of the tested products. The investigators received the “Product S” and “Product P” from the test manager in a blinded state and gave it to the volunteers, thus it was performed in a double-blind fashion. The 24 volunteers were randomly distributed using a block randomization table (e.g., AABB, ABAB, ABBA, BBAA, BABA, and BAAB) such that an equal number of participants were assigned to each group, and the volunteer groups were represented as “Group A” or “Group B”. The volunteers in “Group A” were administered the Product P (test cream) on the right side and the Product S (placebo cream) on the left side of their faces (the area around the eyes). The volunteers in “Group B” were administered the Product S (placebo cream) on the right side and the Product P (test cream) on the left side of their faces. This study has been uploaded to http://cris.nih.go.kr/cris/index.jsp (registration number: KCT0004021, 31 May 2019) from the National Research Institute of Health.

### 2.6. Procedures

In this study, hypothesis and objective are to evaluate the anti-aging effect of 1-MEST on skin wrinkle. In addition, the primary and secondary endpoint are wrinkle parameters and visual assessment of skin wrinkles, respectively. Clinical trial was performed to evaluate whether the test group showed significant difference (*p* < 0.05) in two or more of five wrinkle parameters (primary endpoint) as compared to control group. In addition, if the secondary endpoint showed an effect that could support the primary variable, it was considered to improve aging in skin wrinkles. The volunteers were allocated into “Group A” and “Group B” to conduct a split-face randomized, double-blind placebo-controlled study. Before the study began, the volunteers had a wash-out period of 2 weeks, during which the use of other functional cosmetics was inhibited. During the study period, the participants cleaned their faces using a toner and applied the test products to the area around the eyes twice daily for 12 weeks. They washed the area around the eyes at every visit (baseline, 4 weeks, 8 weeks, and 12 weeks) and entered a room with controlled temperature and humidity (22 ± 2 °C, 50 ± 5%) for 30 min. The primary outcome was assessed by evaluating face wrinkle parameters, while the secondary outcome was evaluated by visual assessment and self-questionnaires were used to assess the efficacy and usability.

### 2.7. Flow of Participants

In this case, 24 women participated in the study. The volunteers were randomized and evenly allocated into two groups with split-face (Group A, n = 12, 24 faces; Group B, n = 12, 24 faces). All volunteers completed the follow-up of this clinical trial without dropping out (Figure 2).

### 2.8. Measurement of the Primary Outcomes

#### Assessment of Wrinkle Parameters Using Skin Replicas

Skin wrinkle parameters were assessed as described in our previous studies [19,20,21]. Briefly, to reflect the skin wrinkles of volunteers, SILFLO replicas were produced using a mixture of resin and catalyst after removing the hair around the test area. The SILFLO skin replica images were obtained using the Skin Visioline^®^ VL650 (C + K, Koln, Germany) by transmitting light at a 35° angle and the green shadows of SILFLO replicas were automatically quantified using the Quantiride^®^ (Monaderm, Monaco). In each evaluation session, the following five parameters of skin wrinkles were assessed: skin roughness (Rt), maximum roughness (Rm), average roughness (Rz), smoothness depth (Rp), and arithmetic average roughness (Ra).

### 2.9. Measurement of the Secondary Outcomes

#### 2.9.1. Evaluation of Visual Assessment of Skin Wrinkles

The detailed method for evaluating the visual Assessment was described previously [19,20,21]. According to the MFDS guidelines, two investigators independently assessed the wrinkles around the participants’ eyes under constant lighting conditions (dayglow color, 820 lm, and 22 W). The grade of the wrinkled state was recorded in 10 stratified steps (increment by 0.5 points) using modified Danielle’s criteria in each assessment. The mean values of wrinkle grades were calculated for each participant and statistically analyzed.

#### 2.9.2. Evaluation of Self-Questionnaires for Efficacy and Usability

A detailed method for evaluating self-questionnaires was reported earlier [19,20,21]. Briefly, the self-questionnaire evaluating the efficacy of test products was completed by all volunteers following each evaluation session. All volunteers completed a self-questionnaire for usability at 12 weeks. The grades for the questionnaire answers were as follows: 1. I disagree at all: it is not good at all; 2. I disagree: it is not good; 3. There is no difference: it is normal; 4. I agree: it is good; 5. I strongly agree: it is excellent. We evaluated positive responses (grades 4 or 5) to both questionnaires.

### 2.10. Analysis of Skin Safety

The method for analyzing skin safety was reported in our previous studies [19,20,21]. In brief, skin safety was analyzed, and the investigators evaluated subjective or objective skin irritation based on questionnaire responses and clinical observation after test product usage.

### 2.11. Statistical Analysis

All data were statistically analyzed using the SPSS ver.20 (IBM, Armonk, NY, USA). The Shapiro-Wilk test and kurtosis and skewness were used to analyze the normality distribution (if kurtosis and skewness were within ±2, data were considered normally distributed), and the preliminary homogeneity test was verified by paired *t*-test. We used repeated measured ANOVA (RM ANOVA) on skin wrinkle parameters to analyze the interdependence (reciprocal action) of repeated measurements and compare the test and control groups or changes between the before and after use. On visual assessment, if the value of intraclass correlation coefficient (ICC) was statistically significant (over 0.8 between two investigators), the reliability between investigators was recognized, and the average value was used for the analysis. Statistical significance was analyzed using the chi-square test and Fisher’s exact test on the self-questionnaires for efficacy and usability. A statistically significant difference was set at *p* < 0.05. The percentage of changes from baseline skin wrinkle parameters and visual assessment was calculated according to the following equation:$$\mathrm{Change}\;\mathrm{rate}\;(\%)=\frac{\mathrm{Measurements}\;\mathrm{after}\;\mathrm{use}\;-\mathrm{Measurements}\;\mathrm{baseline}}{\mathrm{Measurements}\;\mathrm{baseline}}\times 100$$

## 3. Results

### 3.1. 1-MEST Suppresses MMP Expression in HEKs

The chemical structure of 1-MEST is shown in Figure 1. The viability of HEKs was investigated after treatment with 1-MEST from 0.1 to 20 μM; 1-MEST did not induce toxicity in HEKs (Figure 3a). UVB exposure induced the secretion of MMP-1 in HEKs, which was significantly inhibited by 1-MEST treatment in a dose-dependent manner (Figure 3b). Increased MMP-9 protein expression was observed in the HEKs irradiated with UVB, and this effect was significantly suppressed by treatment with 1-MEST (Figure 3c,d).

### 3.2. Baseline Characteristics of Volunteers

The age of the volunteers ranged between 38 to 56 years, and the average age was 49.88 ± 4.34 years. The skin characteristics and conditions were recorded for each participant and evaluated using a questionnaire. The results are presented in Table 2.

### 3.3. Evaluation of Primary Outcomes

#### 3.3.1. Wrinkle Parameters in Skin Replicas of the Volunteers

##### Comparisons of Changes between before and after Use in Each Test and Control Groups

Compared to baseline, the value of Rt significantly decreased in the test group at 4 (* *p* = 0.000), 8 (* *p* = 0.000), and 12 (* *p* = 0.039) weeks. In contrast, in the control group, the Rt value significantly decreased only at 8 weeks (*p* = 0.000; Figure 4a). The decrement rate of Rt was at 12 weeks was 2.17–8.91% in the test group and 0.46–4.10% in the control group (Table 3). The Rm and Rz values of the test group were significantly reduced at 4 (Rm; * *p* = 0.000, Rz; * *p* = 0.011) and 8 (Rm; * *p* = 0.000, Rz; * *p* = 0.000) weeks, whereas the control group showed a significant decrease only at 8 (Rm; * *p* = 0.000, Rz; * *p* = 0.000) weeks (Figure 4b,c). The decrement rate of the test group was 2.13–9.99% for Rm and 2.05–11.78% for Rz; in contrast, the decrement rate of the control group was 0.80–5.08% for Rm and 0.80–6.80% for Rz at 12 weeks (Table 3). Significant decreases in Rp values were observed in the test group at 4 (* *p* = 0.000), 8 (* *p* = 0.000), and 12 (* *p* = 0.027) weeks, but no such significant decrease was observed in the control group after 12 weeks (Figure 4d). The decrement rate for Rp was 2.76–7.62% in the test group and 0.21–1.72% in the control group at 12 weeks (Table 3). Finally, a significant decrease in the Ra value was observed in both the test and control groups at 8 (both; * *p* = 0.000) weeks (Figure 4e), and the decrement rate of Ra was 1.46–15.86% in the test group and 0.10–9.13% in the control group at 12 weeks (Table 3).

##### Comparisons of Changes between Test and Control Groups during Clinical Trial Periods

Compared to the control group, the decrease in Rt and Rm values of the test group was significantly at 4 (Rt; ^†^
*p* = 0.000, Rm; ^†^
*p* = 0.006), 8 (Rt; ^†^
*p* = 0.000, Rm; ^†^
*p* = 0.001), and 12 weeks (Rt; ^†^
*p* = 0.002, Rm; ^†^
*p* = 0.023; Figure 4a,b, and Table 4). Significant differences in the decreased Rz and Ra values were observed in the test group at 8 (Rz; ^†^
*p* = 0.004, Ra; ^†^
*p* = 0.005) and 12 weeks (Rz; ^†^
*p* = 0.049, Ra; ^†^
*p* = 0.036; Figure 4c,e, and Table 4). The decrease in Rp value of the test group was significant at 4 (^†^
*p* = 0.003) and 8 weeks (^†^
*p* = 0.000; Figure 4d, and Table 4).

### 3.4. Evaluation of Secondary Outcomes

#### 3.4.1. Analysis of Skin Wrinkle Using Visual Assessment

The mean visual assessment score significantly decreased in the test group at 8 (* *p* = 0.000) and 12 weeks (* *p* = 0.002) compared to baseline; in contrast, no significant decrease was found in the control group at 12 weeks (Table 5). The decrement rate of visual assessment was 0.85–3.41% in the test group and 0.00–0.35% in the control group at 12 weeks (Table 5). Compared to the control group, the test group showed significant differences in the decreased mean score in visual assessments at 8 (^†^
*p* = 0.003) and 12 weeks (^†^
*p* = 0.027; Table 6).

#### 3.4.2. Analysis of Self-Questionnaires for Efficacy

Among the 24 volunteers, positive responses for the decrease of (fine) wrinkles were recorded in 50.00–83.33% of participants of the test group and 37.50–70.83% of participants of the control group at 12 weeks (Figure 5a). However, no significant differences were observed between the test and control groups (Table 7).

#### 3.4.3. Analysis of Self-Questionnaires for Usability

Among the 24 volunteers, the participants of the test group provided positive responses for “Color” (70.83%), “Scent” (33.3%), “Viscosity” (50.00%), “Absorption” (58.33%), and “Satisfaction” (62.50%) (Figure 4b, left panel). Participants in the control group answered positively for “Color” (79.17%), “Scent” (41.67%), “Viscosity” (41.67%), “Absorption” (58.33%), and “Satisfaction” (66.67%) (Figure 5b, right panel). However, there were no significant differences between the test and control groups (Table 8).

## 4. Discussion

This study aimed to evaluate the safety and efficacy of 1-MEST as an anti-wrinkle agent in a split-face randomized, double-blind placebo-controlled trial. First, we observed that 1-MEST did not show in vitro cytotoxicity in HEKs, and suppressed the effects of MMPs in the HEKs after irradiation with UVB. Second, we found that our test cream containing 0.1% 1-MEST ameliorated wrinkle roughness and depth, as evaluated by primary outcomes. Lastly, we recognized that our test cream had better efficacy than the placebo cream on skin wrinkles evaluated through secondary outcomes and did not cause any adverse events (Appendix A). Thus, the overall results of the present study suggest that 1-MEST is a potential anti-wrinkle agent and can be used to alleviate skin wrinkle formation.

The skin ECM is composed mainly of elastic fibers and collagen, which play an important role in the skin tissue and cells by providing a structural framework. Studies have reported that skin sagging and coarse wrinkle formation occurs through ECM degradation by MMPs [6,7,22]. Notably, the expression of MMPs is induced by prolonged UV light exposure, which causes wrinkle formation due to the breakdown of ECM proteins and collagen [6,7,23]. Therefore, we propose that inhibition of MMP expression is a potential strategy to inhibit wrinkle formation. Based on the above observations, we investigated the secretion and expression of MMPs to determine the anti-wrinkle efficacy of 1-MEST using human primary cells (HEKs). Our results showed that treatment with 1-MEST inhibited the secretion of MMP-1 and the expression of MMP-9 in HEKs. However, we did not elucidate the mechanisms underlying the suppressive effect of 1-MEST on MMP secretion in this study, and therefore, further studies are warranted to clarify these phenomena.

It is well known that UV light is composed UVA, UVB, and UVC and distinguished by wavelength such as UVA having longest (315–400 nm), UVB having mid-range (290–320 nm), and UVC having shortest (100–280 nm). Whereas most UVC is absorbed in atmosphere, UVB can reach epidermis area and UVB can penetrate dermis area [24]. Among the UV light, UVA is considered to play an important role in skin photoaging and UVB is well known as mutagen and inducer of skin cancer. On the other hand, other groups reported that the UVB are mostly responsible for skin changes such as wrinkle formation, epidermal thickening, degradation of matrix macromolecules, vascularization, and immunosuppression. However, UVA is partly absorbed and has lower efficiency in skin damage such as erythema [25,26,27]. In addition, previous studies reported that high-dose UVB irradiation led to apoptotic (e.g., sunburn) [28,29,30] or cancerous phenotypes [31,32] in skin, while low-dose UVB exposure accelerated skin aging or photoaging [1]. Thus, we suggest that both of UVA and low-dose UVB should be considered to evaluate anti-skin aging efficacy. Moreover, given that a skin wrinkle formation is started by structural changes in both of epidermal and dermal layers through ECM degradation due to MMPs expression, not only keratinocytes but also fibroblasts should be investigated on the MMPs expression after exposure UVA or low-dose UVB. In this study, the aim of in vitro study was to investigate whether 1-MEST has a role in the response of keratinocytes to UVB irradiating. In results, 1-MEST inhibited UBV-induced skin damage. In accordance with the in vitro results, anti-wrinkle activity of 1-MEST was clinically proven.

Skin aging is usually classified as intrinsic (chronological) and extrinsic (photoaging) aging. Extrinsic aging shows photoaged skin mainly due to exposure to ultraviolet B and environmental pollutants, while intrinsic aging is related with aging factors including genes, neuroendocrine, and skin diseases. Intrinsic aging can be accelerated by external factors including ultraviolet radiation, pollutants and microbial insults, resulting in a complex biological factory where structural proteins, lipids, neuroendocrine, melatonin and vitamin D are produced to protect skin damage. UV can regulate neuronal signals but chronic exposure to UV induces oxidative stress which leads to neuroendocrine dysregulation and decreased antioxidant defense systems in the skin. Long term of UV absorption by the skin not only decrease skin function but also induces skin pathology (e.g., cancer, aging, autoimmune diseases). The protective abilities to deal with external stressors are regulated by the cutaneous neuroendocrine systems [33,34,35]. These skin functions have interaction between skin’s endocrine system and central nerve system to maintain and cutaneous homeostasis, with UVB being more efficient than UVA. UV radiation makes crosstalk between skin and brain for neuroendocrine system to control body homeostasis by producing neuropeptides, biogenic amines, serotonin, melatonin, acetylcholine, steroids, cytokines, quinones, indoles, and 7-dehydrocholesterol, precursor of vitamin D. In the neuroendocrine aspects of skin aging, most of biologically relevant molecules have chromophores such as aromatic rings corresponding to UV absorption [36,37]. Among them, melatonin is reported as an effective antioxidant hormone to reduce oxidative stress, p53 activation, and NF-κB pathway in radiation-induced premature senescence in terms of indirect way, also it can directly scavenge free radical for skin rejuvenation [38]. Thus, we suggest that further studies would be necessary to regard homeostatic function of the skin or photoprotective and anti-aging properties of melatonin, considering current strategies against skin aging.

In our clinical trial, only women participated, and we could not obtain these results from men. Thus, our results are not applicable to both sexes, and the study should be performed including male participants. Moreover, a larger study cohort with even sex distribution should be included in future studies. Nevertheless, our study outcomes prove that 1-MEST is a potential anti-wrinkle agent. If our findings are validated in future studies, we believe that 1-MEST would be a potential therapeutic agent for cosmetics and skin treatment use.

## 5. Conclusions

In conclusion, this study indicates that treatment with 1-MEST suppresses the expression of MMPs in epidermal keratinocytes. Furthermore, 1-MEST could have therapeutic potential for slowing down or preventing skin aging, such as wrinkle formation, and could be used in cosmetics as an anti-wrinkle agent. Thus, our findings provide useful information for ameliorating skin wrinkle formation and provide scientific evidence in dermatology using microalgae.

## Figures and Tables

**Figure 1 jcm-12-00587-f001:**
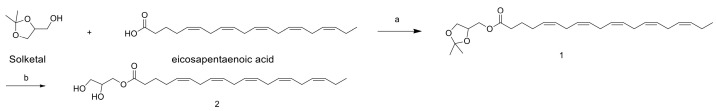
Schematic showing chemical structure of 1-monoeicosapentaenoin. Reagent and condition (a) EDC (1 equiv.), 4-DMAP-dry methylene chloride (0.2 equiv.), RT, 24h. (b) Acetic acid, distilled water, 50 °C, 5 h.

**Figure 2 jcm-12-00587-f002:**
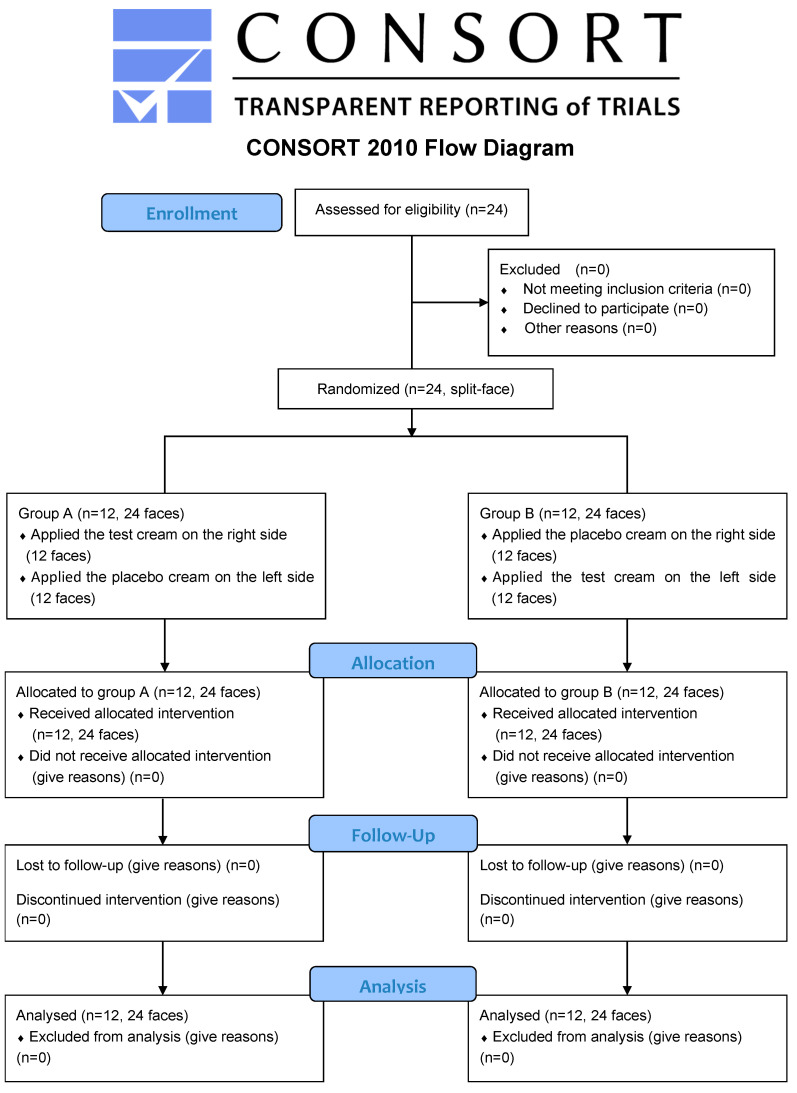
The CONSORT flow diagram of the study participant. This clinical trial adhered to the tenets of the CONSORT statement.

**Figure 3 jcm-12-00587-f003:**
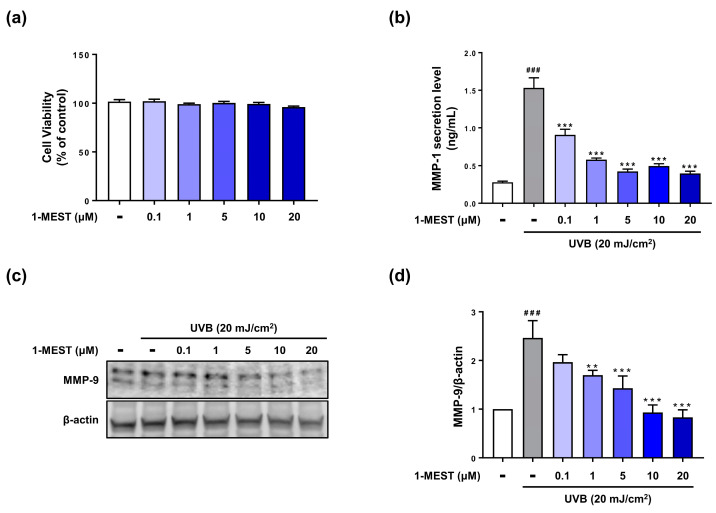
Suppressive effect of 1-MEST on MMP expression in HEKs. (**a**) The viability of HEKs was measured using an MTS assay after 24 h 1-MEST treatment. (**b**) The secretion levels of MMP-1 were investigated by ELISA kit using the culture supernatants of HEKs after 1-MEST treatment and UVB exposure. (**c**,**d**) The protein expression levels of MMP-9 were evaluated using Western blot analysis and quantified by ImageJ software. UVB light; 302 nm for 1 min (20 mJ/cm^2^). The data are presented as the mean ± SD (n = 3). ^###^
*p* < 0.001 vs. normal control. ** *p* < 0.01, *** *p* < 0.001 vs. UVB-irradiated control.

**Figure 4 jcm-12-00587-f004:**
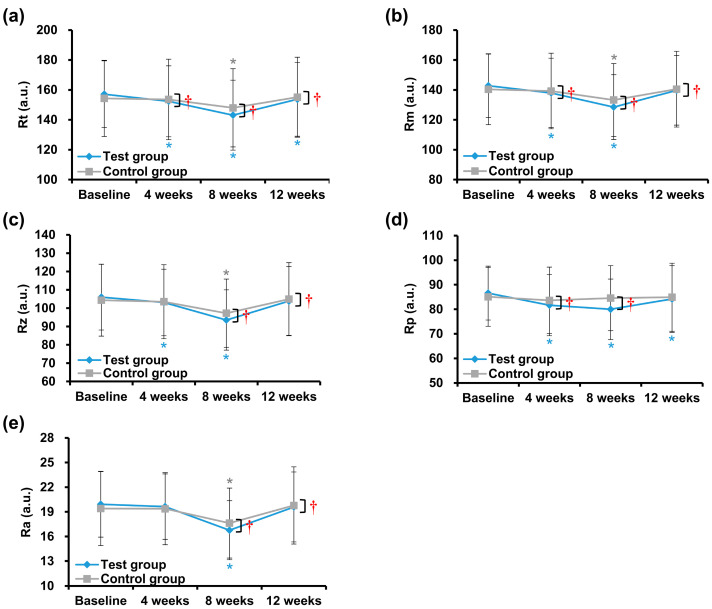
Evaluation of wrinkle parameters using the skin replicas of volunteers. The test cream was applied to the volunteers (test group) and the placebo cream (control group) for 12 consecutive weeks. Changes in (**a**) skin roughness (Rt), (**b**) maximum roughness (Rm), (**c**) average roughness (Rz), (**d**) smoothness depth (Rp), and (**e**) arithmetic average roughness (Ra) were analyzed at baseline, 4, 8, and 12 weeks. Data are presented as the mean ± standard deviation (SD) (n = 24 faces, each group). * *p* < 0.05 after vs. baseline, ^†^
*p* < 0.05 test vs. control group.

**Figure 5 jcm-12-00587-f005:**
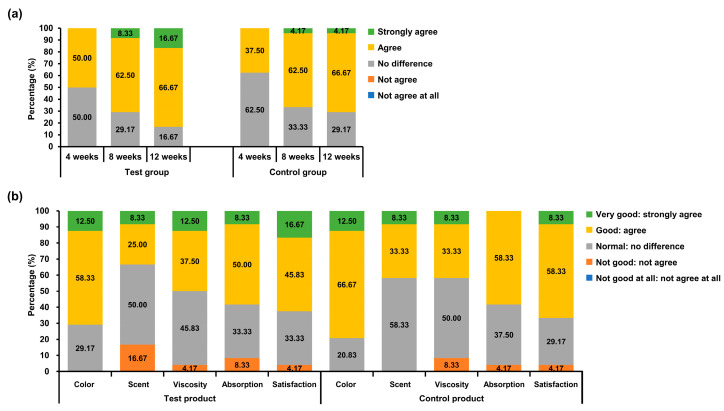
Comparative sensory profiles of the test and control groups for efficacy and usability questionnaires (in terms of positive answers, %). The test group used the cream containing 1-MEST. (**a**) Self-questionnaires for efficacy. (**b**) Self-questionnaires for usability.

**Table 1 jcm-12-00587-t001:** Inclusion and exclusion criteria.

Item	Descriptive Criteria
Inclusion criteria	Adult males and females aged 30 to 65 and who have wrinkles on the test site according to the judgment of the main examiner.
	Healthy persons without major or chronic physical diseases including skin diseases.
	Applicants who voluntarily sign a written consent form after being sufficiently explained about the purpose and content of the test before the test.
	Those who can follow up during the test period.
Exclusion criteria	Pregnant, lactating, or planning to become pregnant within 6 months.
	In the case of using a skin cosmetic product containing steroids for more than 1 month for the treatment of skin diseases.
	Six months have not passed since participating in the same experiment.
	In the case of having sensitive or irritable skin.
	In case of skin abnormalities such as spots, acne, erythema, and expansion of capillaries on the test site.
	If the same or similar cosmetics or medicines are used on the test site within 3 months of starting the test.
	In the case of having a procedure (skin dermabrasion, botox, other skin care, etc.) on the test site or having a plan within 6 months.
	In the case of having chronic diseases (asthma, diabetes, high blood pressure, etc.).
	When the test is judged to be difficult by the main tester’s judgment.

**Table 2 jcm-12-00587-t002:** Information of volunteers’ skin characteristics and condition (n = 24).

Item	Classification	Frequency (N)	Percentage (%)
Age	30’s	1	4.17
	40’s	7	29.17
	50’s	16	66.67
Skin type	Dry	10	41.67
	Normal	8	33.33
	Oily	2	8.33
	Dry and oily	4	16.67
	Problematic	0	0.00
Hydration	Sufficient	0	0.00
	Normal	11	45.83
	Deficient	13	54.17
Sebum	Glossy	2	8.33
	Normal	12	50.00
	Deficient	10	41.67
Surface	Smooth	5	20.83
	Normal	15	62.50
	Rough	4	16.67
Thickness	Thin	10	41.67
	Normal	11	45.83
	Thick	3	12.50
Duration of UV exposure	Less than 1 h	11	45.83
	1–3 h	13	54.17
	More than 3 h	0	0.00
Sleeping hours	Less than 5 h	0	0.00
	5–8 h	23	95.83
	More than 8 h	1	4.17
Smoking	No	24	100.00
	Less than 10 pieces	0	0.00
	More than 10 pieces	0	0.00
Irritability	Yes	0	0.00
	No	24	100.00
Stinging	Yes	0	0.00
	No	24	100.00
Adverse reaction	Yes	0	0.00
	No	24	100.00

**Table 3 jcm-12-00587-t003:** Statistical analysis of skin wrinkle parameters between before and after use in each test and control groups (n = 24).

Parameter	Group	Time Point	N	Mean ^1^	SD	*p*-Value ^2^	Change Rate ^3^ (%)
Rt	Test	Baseline	24	157.13	22.24	-	-
		4 weeks	24	152.40	23.71	0.000 *	3.01▼
		8 weeks	24	143.13	23.40	0.000 *	8.91▼
		12 weeks	24	153.72	24.69	0.039 *	2.17▼
	Control	Baseline	24	154.33	25.45	-	-
		4 weeks	24	153.62	26.88	0.488	0.46▼
		8 weeks	24	148.00	26.09	0.000 *	4.10▼
		12 weeks	24	155.11	26.73	0.426	0.51△
Rm	Test	Baseline	24	142.78	21.34	-	-
		4 weeks	24	137.97	23.22	0.000 *	3.37▼
		8 weeks	24	128.51	21.66	0.000 *	9.99▼
		12 weeks	24	139.74	23.32	0.072	2.13▼
	Control	Baseline	24	140.35	23.63	-	-
		4 weeks	24	139.23	25.29	0.289	0.80▼
		8 weeks	24	133.22	24.36	0.000 *	5.08▼
		12 weeks	24	140.51	25.34	0.876	0.11△
Rz	Test	Baseline	24	106.02	17.97	-	-
		4 weeks	24	103.10	18.16	0.011 *	2.75▼
		8 weeks	24	93.53	16.44	0.000 *	11.78▼
		12 weeks	24	103.85	18.93	0.159	2.05▼
	Control	Baseline	24	104.33	19.54	-	-
		4 weeks	24	103.50	20.16	0.450	0.80▼
		8 weeks	24	97.24	18.60	0.000 *	6.80▼
		12 weeks	24	104.96	19.86	0.453	0.60△
Rp	Test	Baseline	24	86.58	10.97	-	-
		4 weeks	24	81.67	12.49	0.000 *	5.67▼
		8 weeks	24	79.98	12.33	0.000 *	7.62▼
		12 weeks	24	84.19	13.59	0.027 *	2.76▼
	Control	Baseline	24	85.08	11.99	-	-
		4 weeks	24	83.62	13.56	0.148	1.72▼
		8 weeks	24	84.54	13.27	0.670	0.63▼
		12 weeks	24	84.90	13.84	0.867	0.21▼
Ra	Test	Baseline	24	19.92	4.01	-	-
		4 weeks	24	19.63	3.98	0.317	1.46▼
		8 weeks	24	16.76	3.59	0.000 *	15.86▼
		12 weeks	24	19.58	4.26	0.418	1.71▼
	Control	Baseline	24	19.39	4.51	-	-
		4 weeks	24	19.37	4.39	0.967	0.10▼
		8 weeks	24	17.62	4.26	0.000 *	9.13▼
		12 weeks	24	19.77	4.70	0.180	1.96△

^1^ Decrement of the mean-value represents decrease of wrinkle. ^2^ Significantly different at * *p* < 0.05 compared with baseline. ^3^ Change rate, ▼; decrement rate, △; increment rate.

**Table 4 jcm-12-00587-t004:** Statistical analysis of skin wrinkle parameters between test and control groups during clinical trial periods (n = 24).

Group	Parameter	4 Weeks	8 Weeks	12 Weeks
Test vs. Control	Rt	0.000 ^†^	0.000 ^†^	0.002 ^†^
	Rm	0.006 ^†^	0.001 ^†^	0.023 ^†^
	Rz	0.131	0.004 ^†^	0.049 ^†^
	Rp	0.003 ^†^	0.000 ^†^	0.087
	Ra	0.440	0.005 ^†^	0.036 ^†^

^†^ Significantly different at *p* < 0.05 compared with control group.

**Table 5 jcm-12-00587-t005:** Statistical analysis of visual assessment after application in the test and control group (n = 24).

Group	Time Point	N	Mean ^1^	SD	*p*-Value ^2^	Change Rate ^3^ (%)
Test	Baseline	24	5.86	1.03	-	-
	4 weeks	24	5.81	1.03	0.057	0.85▼
	8 weeks	24	5.66	0.97	0.000 *	3.41▼
	12 weeks	24	5.72	1.03	0.002 *	2.39▼
Control	Baseline	24	5.77	1.13	-	-
	4 weeks	24	5.78	1.13	0.575	0.17△
	8 weeks	24	5.75	1.17	0.604	0.35▼
	12 weeks	24	5.77	1.23	1.000	0.00 (-)

^1^ Decrement of the mean-value represents decrease of wrinkle. ^2^ Significantly different at * *p* < 0.05 compared with baseline. ^3^ Change rate, ▼; decrement rate, △; increment rate.

**Table 6 jcm-12-00587-t006:** Statistical analysis of visual assessment between test and control groups (n = 24).

Group	4 Weeks	8 Weeks	12 Weeks
Test vs. Control	0.056	0.003 ^†^	0.027 ^†^

^†^ Significantly different at *p* < 0.05 compared with control group.

**Table 7 jcm-12-00587-t007:** The results of positive answers in self-questionnaires for efficacy (n = 24).

Item	Time Point	Test Group		Control Group		*p*-Value
		N ^1^	% ^2^	N ^1^	% ^2^	
Decrease of (fine) wrinkles	4 weeks	12	50.00	9	37.50	0.561
	8 weeks	17	70.83	16	66.67	0.819
	12 weeks	20	83.33	17	70.83	0.270

^1^ N (Frequency) = Number of positive answers (4, Agree; ~5, Strongly agree). ^2^ % (Percentage) = Number of positive answers/Total number of subjects (24) × 100.

**Table 8 jcm-12-00587-t008:** The result of positive answers in self-questionnaires for usability (n = 24).

Item	Test Product		Control Product		*p*-Value
	N ^1^	% ^2^	N ^1^	% ^2^	
Color	17	70.83	19	79.17	0.792
Scent	8	33.33	10	41.67	0.218
Viscosity	12	50.00	10	41.67	0.888
Absorption	14	58.33	14	58.33	0.467
Satisfaction	15	62.50	16	66.67	0.779

^1^ N (Frequency) = Number of positive answers (4, Good: Agree; ~5, Excellent: Strongly agree). ^2^ % (Percentage) = Number of positive answers/Total number of subjects (24) × 100.

## Data Availability

The data used to support the findings of this study are available from the corresponding author upon reasonable request.

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
