# Peer review of "The Safety and Efficacy of 1-Monoeicosapentaenoin Isolated from the Trebouxiophyceae Micractinium on Anti-Wrinkle: A Split-Face Randomized, Double-Blind Placebo-Controlled Clinical Study"

_jcm, 2023, doi:10.3390/jcm12020587_

Round 1
Reviewer 1 Report
Clinical trial on the potential anti-wrinckle effects of 1-MEST, including some data on the production of the substance and evaluation of its toxicity and some mechanisms.
The study includes a clinical trial with and interesting methodology for the study of effect on face wrinckles, though some relevant issues should be solved before being considered for publication.
.- pg 4, please clarify the block randomization method, as the options are identified as S or P, while given every subject receives both treatments it is in fact group 1 or 2 (side of the test product aplication) what is being randomized.
.- the main variable, hipothesis and objetive should be clearly stated prospectively, as well as the difference (delta) between the groups effect to demonstrate differences (not only the statistical significance).
.- many statistical analysis have been performed, have you considered correcting the alfa-error to avoid multianalysis bias?
.- The formula used to calculate the changes from baseline (pg 6), considers the absolut change of initial-final situation, but this could confuse if the effect is protective or lesive. Instead the final-initial situation, positive or negative, should be consider.
.- The statistical analyses are mainly devoted to test the within group difference from baseline and its significance at every time-point. While it is unclear the test of the expected effect vs placebo. The analysis should be oriented to the test (repeated measures ANOVA could be ok if requirements are complied) of the difference or ratio effect of the experimental product - the effect of the placebo, at different times, as well as the IC95% of the means of the effect of each group and of the difference between groups.The whole analysis should be reconsidered.
.- Figure 3 is considered not significant, as the information contained can be easily explained, and the diagram included in supplementary information. Instead, a figure representing the evolution of the effect of each group through time could be very helpful to estimate the evolution and contrasting the effects of each group.
.- Table 3 and some others are organized to compare the evolution within each group, while not to test the difference of the effects between experimental and placebo products. Table 4 should consider again the results, means, IC and differences between groups instead of recording the p-values.
.- results on assessment of the acceptability of the products are of secondary interest.
.- discussion should clearly be dedicated to discuss the apropriateness and limitations of the methods, and the interest and concordance of results in contrast to other studies, while the review information on different factors affecting skin ageing could be waived.
.- only 7 up to 40 references are from the last 4 years, while the vast majority are quite old. Please revise and select in the light of its relevance and added value to consider the innovative knowledge generated by your study.
Reviewer 2 Report
Kim et al. evaluated the safety and efficacy of 1-monoeicosapentaenoin on MMP-9 expression in keratinocytes and the clinical efficacy and safety of 0.1% 1-monoeicosapentaenoin cream on wrinkles in a half-side comparison.
The methods are well described, the presentation of the results is concise and clear, and the conclusions are sound. Especially the clinical study is well presented and of good quality in the field of cosmetic research. .
I have only minor suggestions to improve the paper:
Line 75: Janghang habor – should this spell „harbor“ ?
Table 1, last paragraph: When the the test…. Eliminate 1 „the“
3.2. Flow of Participants: this section should be moved to the Methods
Finally, it would be interesting to have some more information on the mode of action of 1-monoeicosapentaenoin as an anti-aging agent beyond MMP-9 inhibition. Could the authors provide some data on the antioxidant capacity of 1-monoeicosapentaenoin in comparison to positive controls such as tocopherol or resveratrol? Does 1-monoeicosapentaenoin possess DNA damage protecting properties, for example reduction of the formation of cyclobutane pyrimidine dimers in keratinocytes? Such data is easy to obtain and would considerably improve the quality oft he paper.
Round 2
Reviewer 1 Report
Thanks for the effort in revising the manuscript. Even though there are still unsolved concerns.
.- it remains unclear the main variable, as it is declared the response upon the results on 2 variables of 5 (any combination?).
.- Bias of multiple analyses from the same data is not solved by repeated measurement tests, but requires alpha-error correction
.- question 4 remains unsolved regardless of the homogeneity, instead it is a matter of the formula used for calculation
.- question 7 remains unsolved, as the effect is to be considered by change vs baseline of the test group compared with the effect in the placebo group. Changes or statistical significance into each group considered alone are non relevant. Thus, Table 3 is not rightly oriented, the significance should not be tested intra group but inter group time vs baselin differences.
.- question 9 (discussion), is still not well exposed, as it maintains considerations about facts that are not into the studied variables (e.g. ultraviolet light, etc)
